# Preclinical Studies on the Safety and Toxicity of Photoditazine in the Antibacterial Photodynamic Therapy of Uropathogenic Bacteria

**DOI:** 10.3390/biomedicines11082283

**Published:** 2023-08-17

**Authors:** Olga Streltsova, Artem Antonyan, Nadezhda Ignatova, Katerina Yunusova, Vadim Elagin, Vladislav Kamensky

**Affiliations:** 1Department of Urology Named after E. V. Shakhov, Privolzhsky Research Medical University, 603005 Nizhny Novgorod, Russia; 5x5x5@inbox.ru; 2Department of Epidemiology, Microbiology and Evidence-Based Medicine, Privolzhsky Research Medical University, 603004 Nizhny Novgorod, Russia; n.i.evteeva@gmail.com; 3Department of Pathological Anatomy, Privolzhsky Research Medical University, 603005 Nizhny Novgorod, Russia; katyayunusova@yandex.ru; 4Institute of Experimental Oncology and Biomedical Technologies, Privolzhsky Research Medical University, 603005 Nizhny Novgorod, Russia; elagin.vadim@gmail.com (V.E.); vlad@ufp.appl.sci-nnov.ru (V.K.); 5Federal Research Center Institute of Applied Physics of the Russian Academy of Sciences, 603950 Nizhny Novgorod, Russia

**Keywords:** antibacterial photodynamic therapy, Photoditazine, kidney, C-reactive protein, creatinine, urea, cystatin C

## Abstract

The ‘dusting’ technique of lithotripsy for the removal of infected urinary calculi and the wide use of drainage after endoscopic surgery may stimulate spreading of multidrug-resistant bacterial strains. Antibacterial photodynamic therapy (PDT) is one promising method for the elimination these strains. The purpose of our study was to evaluate alterations of renal pelvis morphology and renal function in laboratory animals after bactericidal regimens of PDT. Renal pelvises of pigs were filled with Photoditazine and then assessed either by examining the accumulation of Photoditazine in the urothelium or by illumination with a laser at a wavelength of 662 nm. A renal test and a complete blood count was performed to assess a negative effect of the treatment on health. Structural alterations of the kidney tissues were analyzed by histological examination. No photosensitizer fluorescence was detected in the urothelium of the pelvis. Histological study showed that PDT caused minor changes to the urothelium of the renal pelvis but did not affect the underlying connective tissue. No renal function abnormalities were found after PDT. Thus, the study indicates that antibacterial PDT is a safety technique that can complement common antibiotic therapy in the surgical treatment of urolithiasis.

## 1. Introduction

Endoscopic surgery, as widely used in modern urology is characterized by less invasiveness and reduced recovery times in the postoperative period than traditional methods. However, this approach includes some hazards such as the difficulty of sterilizing the endoscopic tools after use because of the complexity of their design, as well as human factors that are present at all stages of manipulation. In addition, increases in pressure in the kidney cavity that occur during drainage and that can induce pyelo-interstitial reflux is another risk factor. According to reports, from 30% [1] to 51% [2] of urinary calculi are infected or have a bacterial origin. Modern lithotripsy approaches are based on breaking kidney stones into small fragments that can be removed/washed out through small diameter accesses. In the case of infected stones, large amount of toxins and bacteria are inevitably released during fragmentation [3]. The most serious postoperative complications that occur against a background of such release of bacteria are systemic inflammatory response syndrome, pyelonephritis and urosepsis. It is known that the incidence of systemic inflammatory response syndrome and urosepsis after endoscopic surgery can reach 27.4% and 7.9%, respectively [4,5,6]. It has also been noted that fever can occur after surgery even when a patient has received a prophylactic antibiotic cover and had a confirmed sterile preoperative urine culture. In this case, infection is caused by bacteria associated with the calculi [7,8,9]. In addition, the widespread occurrence of multidrug-resistant strains making antibiotic prophylaxis inefficient is a common clinical problem [10].

In this regard, the development of approaches that permit prevention of postoperative complications is a topical problem. Photodynamic therapy (PDT) is one promising method. Since the 1980s, PDT has been used in clinical practice for the treatment of oncological diseases. In addition, studies have been undertaken into the possibility of using PDT for the treatment of localized bacterial infections, including those resistant to antibiotics [11]. Currently, experimental and clinical studies on the treatment of purulent wounds and purulent septic complications of the ENT organs using PDT are being carried out [12,13]. The potential use of PDT in dentistry has also been shown [14,15]. Furthermore, antibacterial PDT is a promising technology for the treatment of chronic infections [16,17,18]. A study of the effect of PDT on planktonic forms of bacteria in the urine of patients with urolithiasis was previously published by our research group [19,20]. However, there are no data on the use of photodynamic therapy for the prevention of infectious complications in urology, particularly in the treatment of urolithiasis.

The clinical experience with PDT in urology is limited. Until recently the major role for PDT in urology has been for the diagnosis and treatment of bladder cancer. Some current studies describe PDT application in penile oncology for management of carcinoma in situ, as well as in therapy of upper urinary tract carcinoma and urethra [21] (ClinicalTrials.gov Identifier: NCT03617003). For renal cancer, PDT has previously only been tested using a preclinical model despite its potential application [22].

The present study evaluated alterations of renal pelvis morphology and renal function in laboratory animals to assess the negative effects of previously developed regimens [23] of antibacterial PDT on health.

## 2. Materials and Methods

### 2.1. Animal Care and Surgical Procedures

The study was conducted on seven male Landrace pigs aged 4.5 months. The animals were obtained from a local breeder and were held for a minimum of 7 days for acclimation and observation prior to enrollment on the protocol. The pigs were kept in individual enclosures and received a standard diet. Food was withheld for at least 12 h prior to surgery, while water was available at all times. Premedication was performed by intramuscular injection of 0.04 mL/kg of a mixture of Zoletil^®^ 100 (Virbac Sante Animale, Carros, France) and Xilavet^®^ (Pharmamagist Ltd., Budapest, Hungary) in a ratio of 2:1, respectively. The following manipulations were performed under gas/intravenous anesthesia with endotracheal intubation. After intravenous administration of Propofol (B-Braun, Melsungen, Germany) at 1.5–2 mg/kg, the animals were intubated, anesthesia was maintained with Sevorane («Abbott», Maidenhead, UK) and Zoletil 100 was repeatedly administered. A Fabius CS anesthesia machine was used («Dräger Medical GmbH», Lübeck, Germany).

After induction of general anesthesia, animals were placed in lateral decubitus position. The surgical field was cleaned and disinfected with chlorhexidine gluconate and povidone-iodine. An oblique incision in the hypochondriac region was made for kidney manipulation. A retroperitoneal separation of the kidney and the upper third of the ureter was performed. The ureter was clamped with a soft tourniquet and a catheter was inserted through the incision into the kidney pelvis. After treatment was completed, the laser fiber was withdrawn, the ureter was stitched up using a polyglycolic acid suture (Ethicon, Johnson & Johnson, Cincinnati, OH, USA), followed with layered closure of the incision. Finally, the incision was treated with a topical anti-infective agent and covered with a germicidal sticker. Animals received standard postoperative care including analgesia, antibiotics and routine postoperative evaluations.

### 2.2. Photosensitizer Accumulation and Delivery of PDT

Photoditazine (LLC Veta-Grand, Moscow, Russia), a chlorin e6 dimeglumine-based photosensitizer, was supplied as a dark green liquid in sterile glass vials. The compound was stored at 4 degrees Celsius in fully darkened conditions. All manipulation of the drug occurred at very low light conditions, including darkening of the operating room. A normal saline solution containing 5 mg/mL of Photoditazine and 10% of Triton x-100 was injected into the renal pelvis using the catheter. The kidneys of the first 3 animals were used to assessment of photosensitizer accumulation. Fifteen minutes after drug infusion, the renal pelvis was washed with a sterile normal saline solution followed by kidney removal. The photosensitizer accumulation by the kidney tissue was assessed using a fluorescence imaging device (IVIS, Caliper Life Sciences, Hopkinton, MA, USA). Photoditazine fluorescence was excited in the 620/20 nm range and emissions were detected at 680/20 nm.

The last animals were used for analysis of any structural and functional alterations. Fifteen minutes after drug infusion, a 600-micron optical fiber with a cylindrical diffuser 5 mm in length (LLC Polironic, Moscow, Russia) coupled with a diode medical laser (Latus-K, LLC Atkus, Moscow, Russia) was introduced into the renal pelvis. Continuous wave or pulsed illumination was applied through this with an output laser power of 150 or 300 mW at a wavelength of 662 nm. The total light doses were 90 and 180 Joules. Changes in temperature in the pelvis during PDT were monitored using a portable digital multimeter equipped with a thermocouple.

### 2.3. Blood Chemistry Tests

Blood sampling was performed from a vein located on the outer surface of the auricle of the animals before surgery and on the day of sacrifice. Blood was collected into vacuum tubes (BD Vacutainer^®^, Becton, Dickinson and Co., Franklin Lakes, NJ, USA) containing ethylenediaminetetraacetic acid or sodium citrate. The concentrations of C-reactive protein, creatinine, urea and cystatin C were measured using reagents kits produced by Vital Development Corporation JSC (Saint Petersburg, Russia). Cystatin C, a cysteine protease inhibitor, is a better marker of renal function than creatinine and is less affected by age, gender, muscle mass and ethnicity. Moreover, a complete blood count was also performed.

### 2.4. Histological Examination

Histological examination of the kidney tissues was performed at 3 h, 1 day and 3 days after PDT. The kidneys were collected into 10% neutral buffered formalin. After 48 h, the samples were washed and the pelvis regions were excised from them. Next, the samples were dehydrated in isopropyl alcohol (BioVitrum LLC, Saint Petersburg, Russia) and embedded in paraffin. Sections were cut at a thickness of 5 μm and stained with hematoxylin and eosin to analyze structural alterations.

## 3. Results

All of the seven animals enrolled in the experiments completed treatment and all evaluations prior to sacrifice. At the first stage, the initial volume of the pig kidney pelvis was measured. To achieve this, liquid was injected into the pelvis using a syringe while the pelvic–ureteric junction was clamped. According to the measurements, the normal volume was 1.1 ± 0.1 cm^3^. Therefore, for further studies, the final volume of photosensitizer used was one milliliter. Next, the accumulation of Photoditazine in the kidney tissue was assessed. The distribution of Photoditazine along a longitudinal section of the kidney was analyzed using a fluorescence visualization setup. Figure 1 demonstrates that some photosensitizer fluorescence is localized in the ureter, being associated with the incomplete removal of the Photoditazine solution from the organ. Moreover, fluorescence was detected on the lateral surfaces of the kidney, this being caused by soiling of the sample during preparation for the study. Photosensitizer fluorescence is, however, absent in the region of the image corresponding to the pelvis.

Histological examination showed that the untreated pelvis is lined with epithelium, consisting of 5–6 layers of cells forming the smooth contours of the outer layer. Under the epithelium there is loose connective tissue containing small numbers of thin-walled blood vessels. After exposure of the kidney pelvis to the photosensitizer solution no changes in the state of the lining epithelium were detected (Figure 2).

The next stage of the study was devoted to assessment of the condition of the kidney tissues after photodynamic therapy. Illumination of a pelvis containing Photoditazine solution using a continuous wave laser at 150 mW led to the discovery of the formation of a few foci of loosening areas of desquamation in the surface layer of cells. The intercellular spaces of the surface layer of the urothelium had expanded. However, the cellularity and stratification of the pelvic tissue layers were preserved (Figure 3a). Illumination of a pelvis containing Photoditazine solution using a pulsing laser at 150 mW led to the local expansion of the intercellular spaces of the surface layer of the urothelium (Figure 3b). Increasing the continuous wave laser power up to 300 mW led to the appearance of areas of pronounced cell loosening due to the destruction of intercellular contacts. Moreover, confluent areas of destruction of the surface layers of the urothelium were also present. The thickness of the preserved epithelium was 2–3 rows of cells (Figure 3c). Illumination with a pulsing laser at 300 mW led to the formation of a few small foci of desquamation, as well as to the local erosion of the contours (Figure 3d).

Thus, PDT caused minor changes to the urothelium of the renal pelvis but did not affect the underlying connective tissue. To assess any thermal influence on the tissues, a temperature measurement of the Photoditazine solution in the renal pelvis was performed during laser illumination. The temperature was measured in both irrigated and non-irrigated renal pelvis conditions after continuous wave illumination at 300 mW for 20 min. Irrigation was carried out through a nephrostomy. A bag with liquid was placed at 20 cm above the kidney to achieve an intrapelvic pressure not exceeding 30 cm of water column. During and after illumination, no heating of the liquid in the pelvis was detected under either irrigation or non-irrigation.

The functional condition of the animal kidneys after PDT was assessed by determining the concentrations of creatinine, urea and cystatin C in the blood, as well as performing a complete blood count (Table 1). In tests at 1 day after the treatment, the concentrations of creatinine and urea in the blood were increased by 2 and 3 times, respectively, compared with their levels before treatment. However, the measured values were still within the reference values. It should be noted that the value of cystatin C decreased by 15%. It is known that the concentrations of creatinine and urea are affected by various factors, including violations of water balance and damage to muscle tissue. Probably, these changes are due to damage to the tissue during surgery, as well as a decrease in water intake in the postoperative period. Since the concentration of cystatin C was not affected by these factors, we may conclude that there were no violations of the glomerular filtration rate at 1 day after PDT. The concentrations of procalcitonin and C-reactive protein had not changed by 1 day after surgery either, which indicated the absence of inflammatory complications. However, the erythrocyte sedimentation rate was decreased in the postoperative period, while the concentration of platelets in the blood was increased as a response to surgery. At 3 days after the operation and PDT, the concentrations of creatinine, urea, procalcitonin and C-reactive protein were similar to their respective concentrations before the operation. The value of the cystatin C concentration decreased slightly in the postoperative period. Thus, PDT did not lead to renal function abnormality. However, by 3 days the hematocrit, hemoglobin concentration and the mean volume of erythrocytes were decreased while the erythrocyte sedimentation rate had increased. These changes may be caused by blood loss during the operation.

## 4. Discussion

The rehabilitation period for patients with urolithiasis and the cost of treatment may be significantly prolonged due to infectious and inflammatory complications that can occur in the postoperative period. The source of such complications can be either/both microorganisms associated with the stones and nosocomial infections. Antibiotic therapy is very often inefficient due to global spreading of multidrug resistant microorganisms [19,24]. Antibacterial PDT can be used as an alternative technique to antibiotic therapy. In contrast to antibiotics, PDT has many targets in the bacterial cell and excludes the possibility of the development of resistance [25]. The effectiveness of antibacterial PDT against various types of microorganisms has been shown in previous studies. Antibacterial PDT has also been optimized for killing gram-negative uropathogenic microorganisms and has been tested on infected patients’ urine [23]. In this paper, an integrated study of the safety of the technique that has been developed was carried out on animals. For this reason, an assessment of the accumulation of Photoditazine by the epithelial cells of the renal pelvis was made. Using a fluorescence imaging technique, the photosensitizer was shown not to accumulate in the epithelium during a 10 min exposure. This avoids a risk of significant tissue damage occurring during photodynamic exposure. The traditional PDT is based on intravenous infusion of a photosensitizer followed by light illumination [26,27]. This causes a localized ischemia-reperfusion injury [28,29] or a direct tissue injury [30,31] depending on photosensitizer localization. In contrast to traditional PDT, the developed technique involves a local infusion of the photosensitizer into the renal pelvis. Due to its structural features, the urothelium acts as a barrier that excludes the penetration of various substances, ions and water from the bladder into the tissues [32]. Since the developed technique uses high-power density laser radiation, it was important to assess any change in temperature during PDT. Neither the photosensitizer solution nor the renal pelvis tissues became heated, therefore excluding the risk of thermal damage to the tissues in the organ. To assess the health of the renal pelvis tissues after PDT, a histological examination was performed. The absence of significant damage to the epithelium of the renal pelvis of the animals after illumination with various regimens was probably due to the absence of photosensitizer within the cells. Previously, it was shown that patients who received intravenous photosensitizer developed enterovesical fistulas after PDT [33]. Necrotic tubules, glomerular fibrinoid necrosis, thrombosis of capillary loops, interstitial hemorrhage and lymphocytic infiltrates were revealed after intravenous infusion of WST-09 following interstitial illumination of the lower pole of the kidney [22]. It should be noted that superficial urothelium that had disrupted during PDT regenerated in 4 weeks [34].

To assess the functional state of the kidneys and health of the animals, a biochemical blood analysis was performed. Determination of the glomerular filtration rate values is known to be a common and necessary test for the diagnosis and monitoring of renal dysfunction [35,36]. It is known that cystatin C is produced at a constant rate by all nuclear cells of the body, is freely filtered in the renal glomeruli, then reabsorbed and destroyed in the renal tubules; therefore, any increase in its serum level indicates a decrease in the glomerular filtration rate and the development in renal dysfunction [36]. In this study, it was found that the glomerular filtration rate of the animals’ kidneys was not affected by intraoperative PDT, as the cystatin concentration actually became lower compared to the control. Alterations in the erythrocyte sedimentation rate, hemoglobin concentration and hematocrit detected after PDT may be due to blood loss during surgery, as well as to a decrease in the water balance of the animals.

Thus, the study indicates that antibacterial PDT is a safe technique and may complement common antibiotic therapy in the surgical treatment of urolithiasis. In our opinion, the main advantage of PDT is its simultaneous intraoperative application with laser lithotripsy that will allow an antimicrobial effect to be obtained in a very short period.

## 5. Conclusions

The safety of the bactericidal regimens of PDT as an alternative technique to the use of antibiotics was investigated. Filling the renal pelvis with a photosensitizer did not lead to its accumulation in the urothelium. An insignificant morphological alteration was found in the renal pelvis after local infusion of Photoditazine followed by PDT after the bactericidal regimens. This treatment did not impair renal function. This animal study has confirmed the safety of this technique and the possibility of its use intraoperatively in the surgical treatment of urolithiasis in humans. Further preclinical development is needed before any clinical trials.

## Figures and Tables

**Figure 1 biomedicines-11-02283-f001:**
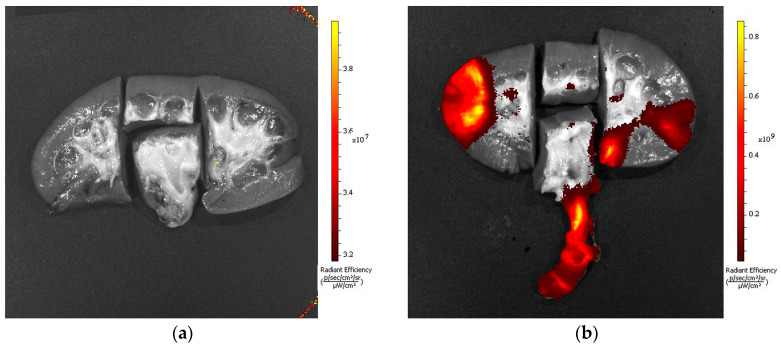
Longitudinal section of an untreated kidney (**a**) and a kidney after 15 min exposure of the pelvis to Photoditazine solution (**b**). Images are macrophotographs with superimposed fluorescence.

**Figure 2 biomedicines-11-02283-f002:**
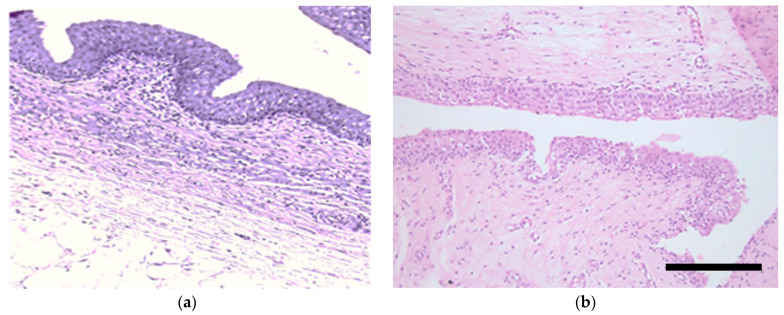
Histology images of the untreated kidney pelvis tissues (**a**) and after 15 min of exposure to Photoditazine solution (**b**). Hematoxylin and eosin staining. Scale bar is 200 µm.

**Figure 3 biomedicines-11-02283-f003:**
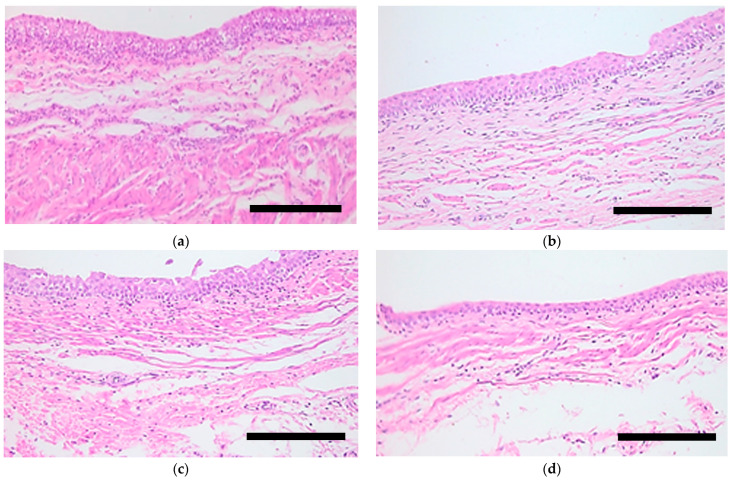
Histology images of kidney pelvis tissues treated by PDT at 150 mW continuous wave (**a**); 150 mW pulsing (**b**); 300 mW continuous wave (**c**); 300 mW pulsing (**d**). Hematoxylin and eosin staining. Scale bar is 200 µm.

**Table 1 biomedicines-11-02283-t001:** Changes in animal blood parameters during PDT.

Blood Parameter	Control Value (Unit)	Animal and Time after PDT			
		Animal 1 0 Day	Animal 1 1 Day	Animal 2 0 Day	Animal 2 3 Day
Creatinine	69.6–207.7 (µM/L)	138.5	139.2	115.2	221.3
Blood urea	3.7–6.4 (mM/L)	4.13	3.79	2.05	7.69
C-reactive protein	- (mg/L)	<0.6	<0.6	<0.6	<0.6
Cystatin C	(mg/L) *	0.44	0.37	0.52	0.37
Procalcitonin	<0.046 (ng/mL)	<0.02	<0.02	<0.02	<0.02
Hemoglobin	99.0–165.0 (g/L)	99.0	100.0	101.0	83.0
Hematocrit	32.0–50.0 (%)	32.8	32.2	33.1	27.0
Erythrocyte sedimentation rate	2.0–9.0 (mm/h)	8.0	3.0	2.0	34.0

* For these animals, there are no reference values for cystatin C; for humans, 0.61–0.95 mg/L.

## Data Availability

The data presented in this study are available on request from the corresponding author.

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
