# Peer review of "Preclinical Studies on the Safety and Toxicity of Photoditazine in the Antibacterial Photodynamic Therapy of Uropathogenic Bacteria"

_biomedicines, 2023, doi:10.3390/biomedicines11082283_

Round 1

Reviewer 1 Report

Only minor alterations to this report are suggested. Line 32: change ‘safety’ to ‘safe’. The authors appear to have insight into the difficulty of sterilizing equipment (line 40). Readers are not told what Photoditazine is, its absorbance spectrum or any other pertinent properties. Use of 662 nm suggests appropriate properties for use in clinical PDT. 

A search of the literature reveals that this agent is a derivative of chlorin e6 with appropriate properties for use as a photosensitizing agent. This should be noted in the report. It would be useful to show an absorbance spectrum.     

Generally good,

Author Response

We would like to thank You for the very useful comments which allow improving the quality of the manuscript. Enclosed below, please, find our detailed reply to the comments.

Only minor alterations to this report are suggested. Line 32: change ‘safety’ to ‘safe’. The authors appear to have insight into the difficulty of sterilizing equipment (line 40).

Suggested corrections have been performed.

Readers are not told what Photoditazine is, its absorbance spectrum or any other pertinent properties. Use of 662 nm suggests appropriate properties for use in clinical PDT.

A search of the literature reveals that this agent is a derivative of chlorin e6 with appropriate properties for use as a photosensitizing agent. This should be noted in the report. It would be useful to show an absorbance spectrum.

Photoditazine, a chlorin e6 dimeglumine-based photosensitizer was used in this study. Information about Photoditazine has been added to Material and Methods section.

Reviewer 2 Report

1. This work is actually two separate works that have relatively little in common - it's quite confusing. 2. “Part I” - retrospective study. This part might even be interesting, but there was practically no statistical test of the collected data (some, few, data were provided with a standard deviation). 3. I see a big methodological problem. If we convert the number of pathogen species per 1000 into the number of cases classified as urinary tract infection, we do not get full numbers. If not every pathogen detected was classified as a urinary tract infection - please provide inclusion or exclusion criteria. 4. The antibiotic susceptibility section contains obvious errors. I will not analyze each, but for example, for E.c. does not perform sensitivity tests for doxycycline, tetracycline or chloramphenicol. See EUCAST guidelines. 5. As I understand it, the animal study was done on two subjects. This does not allow for statistical analysis. 6. The study has no control group.
In conclusion, it is difficult to find a reasonable justification for the publication of this article.

Author Response

We would like to thank You for the very useful comments which allow improving the quality of the manuscript. Enclosed below, please, find our detailed reply to the comments.

  1. This work is actually two separate works that have relatively little in common - it's quite confusing.

This paper actually consists of two parts supplemented each other: a retrospective analysis and an experimental study. The need of studies on experimental animals were reasoned by literature data as well as results of a retrospective analysis of the patient histories in one hospital.

  1. “Part I” - retrospective study. This part might even be interesting, but there was practically no statistical test of the collected data (some, few, data were provided with a standard deviation).

The retrospective analysis was carried out to demonstrate the high workload of the hospital and the increase in the number of patients with infected urine. These results were used as a stimulus for the development of the antibacterial PDT technique.

  1. I see a big methodological problem. If we convert the number of pathogen species per 1000 into the number of cases classified as urinary tract infection, we do not get full numbers. If not every pathogen detected was classified as a urinary tract infection - please provide inclusion or exclusion criteria.

It should be noted that the number of isolated microorganisms is not equal to number of patients with urinary tract infections due to two things. First, two or more species of microorganisms have been isolated from the urine of some patients. Secondly, the distribution of patients between nosological groups were performed according to the main diagnosis. For example, some patients included in "Urolithiasis" or "Oncology" group had positive urine culture for microorganism, but they are not counted in the "UTI" group.

  1. The antibiotic susceptibility section contains obvious errors. I will not analyze each, but for example, for E.c. does not perform sensitivity tests for doxycycline, tetracycline or chloramphenicol. See EUCAST guidelines.

The antibiotic susceptibility was performed according to guidelines of the Ministry of Health of the Russian Federation, which are based on the EUCAST guidelines. The susceptibility of three species of microorganism to the main antibiotics (including doxycycline, tetracycline and chloramphenicol) may be find at the figure 2.

  1. As I understand it, the animal study was done on two subjects. This does not allow for statistical analysis.

The study was performed on 7 male pigs. Corresponding information was added to the materials and methods. Since this is a pilot study the number of animals has been limited.

  1. The study has no control group.

Kidneys without any treatment were served as a control for histological examination. Blood samples collected from the each animals before surgical procedure were used as a control. Corresponding information was added to the materials and methods.

In conclusion, it is difficult to find a reasonable justification for the publication of this article.

The paper has been improved in accordance with Your recommendations.

Reviewer 3 Report

The authors investigated the safety of PDT after the dusting lithotripsy, which seems a new insight into the field. However, there is a lack of presentation especially in the methods section. Therefore, I would recommend this for publication only after addressing the following issues:

The introduction is sufficiently written, however, there is need for some more references such as:

1.       https://www.tandfonline.com/doi/full/10.1080/1040841X.2018.1467876

2.       https://www.worldscientific.com/doi/abs/10.1142/S1088424621300032

3.       https://www.ingentaconnect.com/content/ben/pri/2013/00000008/00000002/art00004

4.       https://pubs.acs.org/doi/full/10.1021/acssuschemeng.7b01501

5.       https://link.springer.com/protocol/10.1007/978-1-60761-697-9_12

6.       https://pubs.rsc.org/en/content/articlehtml/2004/pp/b311900a

7.       https://pubs.acs.org/doi/full/10.1021/acsbiomaterials.7b00751

Materials and Method:

Materials section is missing: Write the name of the materials that were used in this study with their vendor’s name.

Section 2.1: the section is incomplete. A detailed method needs to be incorporated such as, how many subjects were taken, and how the procedure was done.

Section 2.4: Mention the PS name here line no 108 “ renal pelvis 15 minutes after filling it with photosensitizer” What photosensitizer? What was the concentration and formulation?

Section 2.4: What was the irradiation time of the laser in pulsed or continuous illumination? How much was the incubation time with photosensitizer before laser irradiation? The procedure must be clearly defined.

Section 2.5: What was the procedure for blood chemistry tests? What instruments were used?

Section 2.6: the method is incomplete. Write how the histological sections were cut,? What instrument was used? Is there any kit was used for hematoxylin/eosin staining or what concentrations were used? Which microscope was used?

Results and Discussion:

Section 3.1: Again the total number of patients with gender is missing and must be added in the results.

Table 1: there is no separation between genders why?

I am wondering what strategy was used to treat the patients as prophylactic or curing the infection after the endoscopic or open surgery. How was the outcome? Did patients suffer from severe infections after the procedure?

Did the treated animals divide into male/female groups?

Why, other staining techniques such as IHC was not performed? This may provide the inflammatory or other immune responses of the treatment which is also a great concern of any treatment strategy.

Author Response

We would like to thank You for the very useful comments which allow improving the quality of the manuscript. Enclosed below, please, find our detailed reply to the comments.

The authors investigated the safety of PDT after the dusting lithotripsy, which seems a new insight into the field. However, there is a lack of presentation especially in the methods section. Therefore, I would recommend this for publication only after addressing the following issues:

The introduction is sufficiently written, however, there is need for some more references such as:

  1. https://www.tandfonline.com/doi/full/10.1080/1040841X.2018.1467876
  2. https://www.worldscientific.com/doi/abs/10.1142/S1088424621300032
  3. https://www.ingentaconnect.com/content/ben/pri/2013/00000008/00000002/art00004
  4. https://pubs.acs.org/doi/full/10.1021/acssuschemeng.7b01501
  5. https://link.springer.com/protocol/10.1007/978-1-60761-697-9_12
  6. https://pubs.rsc.org/en/content/articlehtml/2004/pp/b311900a
  7. https://pubs.acs.org/doi/full/10.1021/acsbiomaterials.7b00751

The introduction has been improved by suggested articles.

Materials and Method:

Materials section is missing: Write the name of the materials that were used in this study with their vendor’s name.

The section “2.1. Materials” has been added to manuscript.

Section 2.1: the section is incomplete. A detailed method needs to be incorporated such as, how many subjects were taken, and how the procedure was done.

The missing information have been added.

Section 2.4: Mention the PS name here line no 108 “ renal pelvis 15 minutes after filling it with photosensitizer” What photosensitizer? What was the concentration and formulation?

The name of photosensitizer has been added to this sentence. The using concentration of Photoditazine is described in the section above.

Section 2.4: What was the irradiation time of the laser in pulsed or continuous illumination? How much was the incubation time with photosensitizer before laser irradiation? The procedure must be clearly defined.

Irradiation time was 10 minutes for both type of illimination pulsed and continuous. Figure demonstrated scheme of experiments has been added to the manuscript.

Section 2.5: What was the procedure for blood chemistry tests? What instruments were used?

The analysis was carried out using an automated biochemical analyzer BS-120 (Mindray, China) and a hematology analyzer BC-6800 (Mindray, China) and standard reagents in accordance with the manufacturer’s protocol.

Section 2.6: the method is incomplete. Write how the histological sections were cut,? What instrument was used? Is there any kit was used for hematoxylin/eosin staining or what concentrations were used? Which microscope was used?

Description of histological analysis has been added to manuscript.

Results and Discussion:

Section 3.1: Again the total number of patients with gender is missing and must be added in the results.

The total number of patients has been added. The gender specificity did not analyzed in context of this study.

Table 1: there is no separation between genders why?

The gender specificity did not analyzed in context of this study. This table shows the large amount of work carried out in the hospital.

I am wondering what strategy was used to treat the patients as prophylactic or curing the infection after the endoscopic or open surgery. How was the outcome? Did patients suffer from severe infections after the procedure?

The outcome of curing the infection after the surgery go beyond the scope of this paper. Previously published article contains sample data on this theme. Streltsova O.S., Vlasov V.V., Grebenkin E.V., Antonyan A.E., Elagin V.V., Lazukin V.F., Ignatova N.I., Kamensky V.A. Controlled fragmentation of urinary stones as a method of preventing inflammatory infections in the treatment of urolithiasis (experience in successful clinical use). Sovremennye tehnologii v medicine 2021; 13(3): 55–63, https://doi.org/10.17691/stm2021.13.3.07

Did the treated animals divide into male/female groups?

The study was conducted on seven male pigs.

Why, other staining techniques such as IHC was not performed? This may provide the inflammatory or other immune responses of the treatment which is also a great concern of any treatment strategy.

Inflammatory or immune responses are accompanied by immigration of neutrophils, lymphocytes and monocytes into damage tissue area. These cells are clearly seen on H&E stained sections. It is no need for IHC staining to detect tissue inflammation.

Round 2

Reviewer 2 Report

In reviewer opinion Authors have not addressed all issue pointed out in review

Author Response

Dear Reviewer,

We have taken into account your recommendations regarding the article conception. The part of retrospective analysis was excluded. The section of animal study has been revised in order to make it more understandable and more accurately reflecting of what has been done.

Sincerely yours,

Vadim Elagin

(on behalf of the authors team)

Round 3

Reviewer 2 Report

Thank you for taking into account some of my comments, which certainly made the text more coherent and easier to understand.

In the current version, I don't see a part of the study that correlates with the antimicrobial effect of the intervention.

I believe a more appropriate title would be - "Preclinical Studies on the Safety and Toxicity of Photoditazine in the Treatment of XYZ". This is not a suggestion of a specific title, but an indication of direction.

Author Response

Dear Reviewer,

Thank you for your help to improve our manuscript. We have corrected the title according to your suggestion. 

The revised title is "Preclinical Studies on the Safety and Toxicity of Photoditazine in the antibacterial photodynamic therapy of uropathogenic bacteria".

Sincerely yours,

Vadim Elagin

(on behalf of the authors team)